# Research Progress on Low-Surface-Energy Antifouling Coatings for Ship Hulls: A Review

**DOI:** 10.3390/biomimetics8060502

**Published:** 2023-10-21

**Authors:** Zhimin Cao, Pan Cao

**Affiliations:** 1Institute of Intelligent Manufacturing and Smart Transportation, Suzhou City University, Suzhou 215104, China; 2College of mechanical Engineering, Yangzhou University, Yangzhou 225127, China

**Keywords:** marine biofouling, low surface energy, surface antifouling, surface microstructure, surface modification

## Abstract

The adhesion of marine-fouling organisms to ships significantly increases the hull surface resistance and expedites hull material corrosion. This review delves into the marine biofouling mechanism on marine material surfaces, analyzing the fouling organism adhesion process on hull surfaces and common desorption methods. It highlights the crucial role played by surface energy in antifouling and drag reduction on hulls. The paper primarily concentrates on low-surface-energy antifouling coatings, such as organic silicon and organic fluorine, for ship hull antifouling and drag reduction. Furthermore, it explores the antifouling mechanisms of silicon-based and fluorine-based low-surface-energy antifouling coatings, elucidating their respective advantages and limitations in real-world applications. This review also investigates the antifouling effectiveness of bionic microstructures based on the self-cleaning abilities of natural organisms. It provides a thorough analysis of antifouling and drag reduction theories and preparation methods linked to marine organism surface microstructures, while also clarifying the relationship between microstructure surface antifouling and surface hydrophobicity. Furthermore, it reviews the impact of antibacterial agents, especially antibacterial peptides, on fouling organisms’ adhesion to substrate surfaces and compares the differing effects of surface structure and substances on ship surface antifouling. The paper outlines the potential applications and future directions for low-surface-energy antifouling coating technology.

## 1. Introduction

Ship hull surfaces are prone to biofouling, resulting from the accumulation of fouling organisms upon entering the marine environment. This is primarily due to the marine environment being a complex ecosystem housing a myriad of organisms, including various algae, shellfish, bacteria, and microorganisms [1]. Marine fouling organisms encompass animals, plants, and microorganisms that thrive on the surfaces of ships and oceanic facilities. When vessels venture into the ocean, an organic layer rich in fats, polysaccharides, and proteins accumulates on the surface within minutes [2]. Over the subsequent hours, microorganisms like algae and bacteria proliferate on this organic layer, giving rise to the phenomenon of marine fouling (Figure 1). Consequently, extracellular proteins (EPS) with strong adhesive properties are secreted, forming a biofilm consisting of water, organic matter, microorganisms, and their extracellular by-products. The formation of this biofilm further enhances the adhesion and growth of barnacles, shellfish, and various macroscopic fouling organisms on ship hulls or marine structures [3,4]. These fouling organisms adhere to the ship’s surface, leading to material corrosion. The activities of sulfate-reducing bacteria and iron bacteria expedite chemical corrosion rates, progressively damaging the metal surface coating and maximizing the contact between the fouling organisms and the metal surface. Algae and phytoplankton produce oxygen via photosynthesis, accelerating metal corrosion and diminishing the ship’s operation reliability. The addition of fouling organisms significantly increases the sample’s weight and surface roughness. Substantial fouling organism attachment can notably augment a ship’s weight. Without effective countermeasures, the biological fouling mass generated per square meter of the ship can reach up to 150 kg within six months. Large ship hulls typically have underwater surface areas exceeding 40,000 square meters, and the total attached fouling organisms mass can amount to 6000 tons. Fouling organism adhesion induces alterations in the ship’s surface morphology, resulting in a marked increase in surface roughness. Previous research indicates that when the ship’s biological fouling rate reaches 5%, the friction coefficient rises by 50%, leading to a 40–50% increase in fuel consumption [5,6].

To mitigate the deleterious impact of marine fouling organisms and avert their adverse effects on ships and marine structures, extensive research efforts have been undertaken by scholars to achieve ship fouling prevention and drag reduction [7]. In instances where a substantial quantity of fouling organisms adheres to a vessel’s surface, remediation often necessitates docking and employing physical antifouling techniques, such as mechanical stripping, cavitation water jet cleaning, coating antifouling, ultrasonic antifouling, and radiation. These physical antifouling methods are labor-intensive, costly, and not suited to modern ship fouling prevention and drag reduction strategies [8]. Chemical antifouling methods, on the other hand, prevent fouling by eliminating fouling organisms via the application of chemical agents, including antibiotics, biocides, buffering agents, and preservatives. Historically, prior to the 1980s, the majority of ship surfaces were coated with layers containing heavy metals, like copper, mercury, lead, or organic tin compounds. These heavy metals and organic tin derivatives effectively deterred fouling organisms by impeding their adhesion to ship surfaces, thus mitigating fouling issues [9,10]. Nonetheless, these heavy metals and organic tin substances could enter the human food chain via the bioaccumulation of marine organisms like fish. Consequently, the application of organic tin compounds in antifouling coatings was completely banned by the International Maritime Organization (IMO) [11]. As a result, there is a pressing need for research on environmentally friendly materials and novel methodologies for antifouling and drag reduction.

Surface energy is a pivotal factor influencing the antifouling efficacy of ships [12], and surfaces characterized by a low surface energy can deter the initial attachment of benthic organisms. This inherent low surface energy level hinders fouling organisms’ initial adhesion, and any adherent organisms are more likely to detach from the ship’s surface due to their own gravity and the erosive action of seawater. This effect substantially diminishes corrosion and the burden of fouling organisms [13,14]. The Baier curve depicts the relationship between relative adhesion force and surface energy, showing that the adhesion force first decreases and then increases with the increasing surface energy, reaching its minimum at 22–24 mJ/m^2^. At this point, the surface adhesion force is minimal, rendering the surface less prone to fouling organisms’ attachment [15,16]. The exploration of low-surface-energy hull materials or coatings holds significant practical importance. Employing hull coatings serves as a convenient and cost-effective measure for ship pollution prevention and drag reduction [17]. Unlike traditional antifouling coatings, low-surface-energy antifouling coatings are devoid of toxic components, align with environmental requisites, and offer long-lasting antifouling benefits. Hence, low-surface-energy coatings have emerged as a primary focus in contemporary ship antifouling coating research. From both economic and technological standpoints, fluoride-based low-surface-energy coatings and organosilicon-based antifouling and drag-reducing materials currently represent the most investigated low-surface-energy coatings in antifouling and drag reduction research [18].

## 2. Silicone-Based Materials

Extensive research has been undertaken on organic silicone resin materials, primarily due to silicone-based coatings’ exceptionally low surface energy. Organosilicon resin typically pertains to highly branched organic polysiloxanes which are usually derived from the hydrolysis and condensation of organic silicon monomers. These materials encompass the advantages of both inorganic and organic substances. The high-level bond energy and substantial bond angle associated with Si-O bonds, in conjunction with the pliant Si-O-Si backbone, contribute to the side-chain groups’ protective influence on the main chain, resulting in reduced surface energy. The initial patent for organic silicone antifouling paint was granted in the United States in 1972, which was made of silicone rubber. However, the paint’s application was limited to certain offshore aquaculture industries, primarily due to its extended curing period and inadequate adhesion quality.

Chemical modification is imperative to address the inadequate adhesion of silicone resin to substrates. This is achieved via the reaction of hydroxyl groups on the polysiloxane chain with active groups in various compounds or polymers, leading to the development of resin materials with diverse performance characteristics. For example, antifouling coatings employing acrylic siloxane, siloxane and vinyl polymer, silicone-modified epoxy resin, and amino acid siloxane, epoxy, or polyurethane-modified silicone have demonstrated exceptional antifouling capabilities [19]. The limited durability of superhydrophobic coatings has imposed constraints on their practical application in fields such as marine and construction. Composite coatings composed of epoxy resin (EP), polydimethylsiloxane (PDMS), and SiO_2_ nanoparticles were fabricated using room-temperature liquid spraying and laser curing techniques. The coatings exhibited robust fouling and corrosion resistance even after being immersed in artificial seawater for 72 h. Remarkably, they retained their superhydrophobic properties even after undergoing 100 cycles of load-induced friction [20]. Zhang et al. [21] introduced a silicone-based low-surface-energy antifouling coating incorporating functional hydroxyl siloxanes and crosslinking agents, which exhibited remarkable resistance to marine bacteria, biofilm, and diatom adhesion. Hu et al. [22] formulated hydrophilic fouling-release coatings, which exhibited robust stability and facilitated the detachment of fouling organisms under the influence of water and gravity. Silicone resin-based self-polishing antifouling coatings combine the hydrolysis attributes of self-polishing antifouling coatings with the low-surface-energy characteristics of siloxane resin coatings. These materials offer strong adhesion, stability, and excellent antifouling performances [23]. The Intersleek 700 coating developed by Akzo Nobel in the Netherlands leverages organic silicon technology to maintain a smooth ship surface that is less susceptible to fouling organisms, ensuring easy detachment even with minimal adhesion [24].

Furthermore, the enhancement of corrosion resistance in epoxy coatings is achievable via the incorporation of a porous nanocontainer derived from a UiO-66-NH_2_/CNTs nanocomposite characterized by exceptional barrier properties [25]. Ji et al. [26] introduced a pH-responsive, self-healing, and non-toxic anticorrosion coating based on core−shell nanofiber containers, which exhibited self-healing efficiencies of up to 95.96% and 97.04% in alkaline and acidic solutions, respectively. Dai et al. [27] developed a novel hydrolysis-induced zwitterionic monomer copolymerized with methyl methacrylate (MMA). This silicon-based copolymer rapidly generates zwitterionic surfaces with antifouling properties in marine environments. In the quest for high-strength, two-component coatings with a low surface energy, organic siloxanes are also subjected to modification. Comprising hydroxyl-terminated polydimethylsiloxane, polyurethane, and their respective curing agents, accelerators, and solvents, this modified silicone resin significantly mitigates the limitations of pure silicone resin and silicone rubber. Nevertheless, the introduction of numerous non-low-surface-energy components on the molecule has a substantial impact on its surface energy [18]. Silicone polymer-based coatings characterized by a low surface energy and elastic modulus have shown effectiveness in inhibiting biofouling. Nonetheless, their limitations, including non-repairable properties and a suboptimal antifouling performance under static conditions, constrain their practical applications. In order to address the inadequate mechanical properties of PDMS, which are often insufficient for numerous marine applications, Qiu et al. [28] introduced PDMS-polythiourethane (PTU) composites reinforced with tetrapodal-shaped micro-nano ZnO particles. PTU, characterized by its hydrophilic nature, UV stability, and resistance to biocorrosion, serves as a mechanically robust matrix polymer, making it an excellent candidate for composite materials tailored to maritime applications. Liu et al. [29] introduced a self-repairing coating composed of PDMS-PUa and a minor quantity of the organic antifoulant DCOIT. The coating exhibits the complete recovery of its mechanical properties following damage, whether in air or artificial seawater at room temperature, with accelerated recovery observed at higher temperatures. Additionally, the release rate of DCOIT remains nearly constant and can be controlled via its concentration. Six-month marine field tests confirmed the system’s excellent antifouling/fouling release performance, even in static conditions. They also reported a novel poly(dimethylsiloxane) (PDMS) coating crosslinked via coordination bonds formed between the 2-(2-benzimidazolyl)ethanethiol (BET) and zinc ions [30], and this modified PDMS maintained a low surface energy while demonstrating enhanced adhesion strength when compared to that of the PDMS elastomer. The metal coordination renders its crosslinking process reversible, enabling effective self-healing in both air and artificial seawater at room temperature. Furthermore, the incorporation of zinc–imidazole complexes, which possess antifouling properties, significantly bolsters the fouling resistance against marine bacteria and diatoms.

## 3. Fluorine-Based Materials

Fluorinated resin encompasses polymer materials that incorporate fluorine atoms into the carbon atoms of either the main or side chains. This category includes fluoroolefin polymers and copolymers of fluoroolefins in conjunction with other monomers. Organic fluorine compounds possess fluorine atoms characterized by high-level electronegativity, small atomic radii, short C-F bonds, and an elevated bond energy. These fluorine atoms effectively envelop and occupy the space between the C-C bonds, preventing any atoms or groups from entering, and thus, preserving the integrity of the C-C bonds. Consequently, fluorine-containing resins exhibit robust hydrophobicity and exceptional chemical inertness [31]. Fluorine resin is harnessed in low-surface-energy antifouling coatings to diminish the adhesion and fouling of organic matter by enhancing the substrate surface tension. However, despite their low surface energy, pure organic fluorine resins suffer from restricted molecular mobility, impeding the left and right rotations of the molecules. As a result, these materials exhibit a high bulk elastic modulus and necessitate substantial critical stress to disrupt the adhesive–substrate interaction, rendering the removal of adhered organisms challenging.

Fluororesins were initially investigated to fulfill the evolving requirements of the military industry. In 1938, Dr. Plunket serendipitously uncovered the spontaneous polymerization of tetrafluoroethylene at room temperature, yielding a white powder. This breakthrough prompted a cascade of research endeavors dedicated to the application of fluororesins [32,33]. The Intersleek900 fluoropolymer non-stick fouling coating is suitable for all vessels traveling at speeds exceeding 5.1 m/s. This innovative coating boasts an exceptionally smooth hull surface, ultra-low average hull roughness, outstanding non-stick fouling properties, and a static antifouling performance. Intersleek^®^ 1100SR introduced a pioneering fluorine-free stain-release technology devoid of fungicides, effectively resolving ship fouling adhesion issues. As a result, it secured the 2014 Clean Transportation Maritime Award [34]. In order to leverage the low surface energy of organic fluorocarbon resins for antifouling coatings, traditional fluorocarbon resins have undergone modifications. Three distinct approaches have been developed for creating high-performance, fluorinated, low-surface-energy antifouling coatings. These strategies encompass introducing fluoride as a filler in combination with other resins to produce advanced coatings, incorporating fluoride surfactants into polymers, employing a fusion of fluoropolymers, and selecting large-monomer polymers to formulate fluorinated resin coatings. For example, a fluorine-based coating made of perfluoroalkyl polyether polyurethane, with 10 μm particle-sized polytetrafluoroethylene powder employed as a filler, has been deployed on the US Navy’s “Parrot” vessel. However, it necessitates bi-annual dry-dock cleaning, hindering its widespread utilization for ship antifouling purposes. Cheng et al. [35] introduced a dual-functional low-surface-energy coating with antifouling and anticorrosion attributes via the integration of crosslinking polysilazane preceramic precursor infused with fluorine. The surface free energy (SFE) of this coating is less than 30 mJ/m^2^ and it demonstrates additional antibacterial and antifouling capabilities. Incorporating a perfluorinated surfactant into a liquid polymer containing polar groups results in the surfactant forming a fixed monolayer on the polymer’s surface during the curing process. This integration substantially reduces the critical surface tension of the matrix material and confers remarkably low-surface-energy properties to the paint films. Nevertheless, one significant drawback of such paint films is their rapid surface energy elevation when exposed to water, rendering them ineffective in resisting fouling. This issue arises from the loss of surfactants and molecular rearrangement driven by microorganisms. To maintain a low surface energy on the paint film’s surface in the long term, it is imperative to immobilize the groups with low-surface-energy characteristics. Song et al. [36] reported the successful synthesis of a copolymer, PBAF, using a free radical polymerization method incorporating borneol monomers and fluorine. This coating demonstrates remarkable antibacterial and antifouling performances with resistance levels of 98.2% against *Escherichia coli* and 92.3% against *Staphylococcus aureus*. This exceptional antifouling property is attributed to the incorporation of fluorine components into the copolymer, and this results in a reduced surface energy and enhanced hydrophobicity; importantly, the PBAF coating is environmentally friendly and stable in the long term. Furthermore, Sun et al. [37] harnessed a grafting strategy that combines Atom Transfer Radical Polymerization (ATRP) and Reversible Addition–Fragmentation Chain Transfer (RAFT) in polymerization reactions to produce a range of fluorinated amphiphilic asymmetric polymer brushes featuring semi-fluorinated hydrophobic PPTFMA and hydrophilic PEG side chains. These polymer coatings exhibit a robust antifouling performance. Guo et al. [38] prepared a novel amphiphilic block copolymer PVP–PFA–PDMS combined with fluorine-based and silicone-based materials that was blended into a cross-linked PDMS matrix to form a set of controlled surface composition and surface-renewal coatings with efficient antifouling and fouling-release properties. These coatings incorporated the biofouling settlement resistance ability attributed to hydrophilic PVP segments and the reduced adhesion strength attributed to the low surface energy of fluorine–silicon-containing segments. To enhance the long-term stability of fluoropolymers, the fluorinated side chains of the polymer must be able to organize into an ordered structure on the surface while upholding a high density of −CF_3_ groups on the coating surface [39]. Another strategy involves constraining the chain mobility of the polymer via the chemical crosslinking of groups, such as epoxy-functional groups. For instance, Zhu et al. [40] devised and synthesized functional crosslinkable fluoropolymers comprising short fluoroalkyl groups or perfluoropolyether-substituted fluorinated styrene, methyl methacrylate, and glycidyl methacrylate via free radical polymerization.

The reduction in the surface energy is pivotal, even determinative, for organic fluorine antifouling coatings. James’ research has established a hierarchy of surface energy among the various functional groups of polymers, which is −CH_2_ (36 mN m^−1^) > −CH_3_ (30 mN m^−1^) > −CF_2_ (23 mN m^−1^) > −CF_3_ (15 mN m^−1^). Therefore, maximizing the content of—CF_3_ groups is imperative to ensure a sufficiently extensive fluorinated group on the surface, resulting in a high fluorine content, which significantly reduces the surface energy value. Furthermore, organic fluorine antifouling coatings should possess an exceptionally smooth surface and surface-fluorinated groups capable of withstanding biomolecule-induced molecular rearrangement.

It is worth noting that the developmental challenges associated with organic fluorine antifouling coatings are considerably greater than those of organic silicon antifouling coatings. Despite the excellent performance of fluorine resin, the high cost, elevated curing temperature requirements, and limited adhesion between the coating and the substrate have hindered the creation of an ideal commercial product for low-surface-energy antifouling coatings made of organic fluorine resin. Previous studies have sought to overcome these challenges by developing a novel type of low-surface-energy antifouling coating, combining the affordability of organic silicon, the low surface energy of organic fluorine, and the robust mechanical properties offered by both. This innovative coating is constructed using fluoropolysiloxane as the base material. This fundamental principle involves employing a siloxane chain as the primary structure while introducing a specific concentration of −CF_3_ groups within the side chain. These groups exhibit a strong surface affinity, ensuring strict surface orientation. Consequently, the overall macromolecule maintains the high elasticity and excellent fluidity characteristic of linear polysiloxane while capitalizing on the ultra-low-surface-energy features of −CF_3_ groups [39]. In essence, silicon fluorine resin low-surface-energy antifouling coatings seamlessly amalgamate the desirable attributes of both organic silicon and organic fluorine antifouling coatings, thus delivering an exceptional antifouling performance. Despite the demonstrated effectiveness of this coating, its intricate preparation process and relatively high cost have, to some extent, limited its widespread adoption in antifouling and drag reduction applications.

## 4. Low-Surface-Energy Materials Based on the Surface Microstructure of Marine Organisms

The ocean is home to a multitude of organisms equipped with remarkable antifouling capabilities, indicating the existence of unique substances or surface structures that deter fouling organisms from adhering. This observation has prompted scientists to consider the biomimetic replication of surface microstructures in marine organisms or the extraction of surface secretions. It highlights the opportunity to explore the anti-adhesion mechanisms of marine organisms from the perspectives of marine biology and chemical ecology.

For example, sharks can rapidly and efficiently swim, which is possible due to their skin microstructure. Other researchers have unveiled the shield-shaped scales and strip-like grooves covering the surface of sharks’ skin, making it a challenging substrate for fouling organisms to attach to. This extraordinary antifouling and drag reduction capability has spotlighted biomimetic research on shark skin. The previous experimental results have demonstrated that the imitation shark skin coating can achieve an impressive 85% reduction in fouling organism adhesion [41,42]. Other than sharks, numerous natural organisms exhibit distinctive surface microstructures. Notably, lotus leaves possess a micro/nanostructure and wax-like substances that grant them self-cleaning properties, which is often referred to as the “lotus leaf effect” [43]. Waterfowl feathers, as seen on ducks and geese, exhibit neatly arranged micron and submicron-sized strip structures on their surfaces, providing them with excellent hydrophobicity and breathability [44]. The wings of insects, such as dragonflies and butterflies, are adorned with micrometer-sized overlapping scales, each featuring intricately organized periodically layered nanostriped structures that contribute to their superhydrophobic nature [45]. Shellfish, despite their prolonged immersion in water, remain unaffected by marine fouling organisms. This phenomenon is closely associated with their surface microstructure. Replicating this shell surface microstructure using low-surface-energy material polydimethylsiloxane (PDMS) has been successful. Adhesion experiments on test plates with varying roughness and contact angles have confirmed the antifouling efficacy of this microstructured surface [46,47]. Zhang et al. [48] have precisely designed a highly controllable surface quasi three-dimensional structure manufacturing method based on a two-dimensional laser-induced thermal deformation theory, providing the technical foundation for fabricating antifouling composite scale microstructures. Moreover, the surfaces of marine organisms, such as corals and fish, feature unique microstructures that impart antifouling effects [49]. Experts have become interested in the special microstructures of some biological surfaces, such as the feet of creatures like geckos and spiders, which enable wall-clinging, or water striders, which effortlessly traverse water surfaces [50].

According to Yang’s equation, the contact angle of an object serves as an indicator of its surface energy. Dupre’s derived formula establishes a correlation, wherein a lower solid surface energy level leads to reduced adhesion and, subsequently, an increased contact angle between the solid surface and the liquid. However, minimal adhesion does not always equate to the lowest surface energy level, as the final outcome hinges on the summation of the surface energy and other interactions. The most prevalent and convenient approach for assessing solid surface energy is the contact angle method [51]. This method for calculating the solid surface energy via the contact angle is based on Young’s equation. When the contact angle of a solid surface exceeds 150°, it imparts superhydrophobic properties, significantly diminishing ship fouling. Conversely, when the contact angle falls below 50°, fouling organisms struggle to adhere to the ship’s hull [52]. Ultra-high contact angles of nearly 0° can even be achieved using ultraviolet laser induction, and this technique has found applications in underwater antifouling. Nevertheless, the minimal contact angle is associated with high water resistance, elevating the ship’s drag. Hence, the development of superhydrophobic surfaces based on the contact angle represents a promising strategy to mitigate ship fouling issues [53].

Several methods exist for calculating the solid surface energy based on the contact angle approach, including the Zisman method, Good–Guifalco method, Fowkes method, Wu harmonic averaging method, Owens and Wendt method, LW-AB method, and ZDY method. Nonetheless, there is still ongoing debate surrounding the suitability and applicability of these various methods [54]. Calculating the surface energy based on the droplet contact angle is achieved via the Owens–Wendt–Rabel–Kaelble (OWRK) method, as shown in Formula (1) [55]:(1)$$\left(1+\cos\theta \right)\gamma_{L}=2\left(\sqrt{\gamma_{S}^{LW}\gamma_{L}^{LW}}+\sqrt{\gamma_{S}^{+}\gamma_{L}^{-}}+\sqrt{\gamma_{S}^{-}\gamma_{L}^{+}}\right)$$

Herein, *θ* and $\gamma_{L}$ are the contact angle and solid tension of the liquid, respectively; $\gamma_{S}^{LW}$ and $\gamma_{L}^{LW}$ are solid and liquid Lifshitz–van der Waals components, respectively; $\gamma_{S}^{+}$ and $\gamma_{L}^{+}$ are the Lewis acidic components of the solid and liquid, respectively; $\gamma_{S}^{-}$ and $\gamma_{L}^{-}$ are the Lewis alkaline components of the solid.

Utilizing the established surface parameters of frequently employed detection liquids, a judicious selection of combinations enables the estimation of the sample’s surface energy. Namen et al. [56] determined the surface energy and wettability of composite materials by employing two distinct titration systems. In a separate study, Çıtak et al. [55] assessed the surface energy of samples following the deposition of amorphous carbon onto polyethylene terephthalate by employing three different liquid titration systems. Han et al. [54] obtained the surface energy of various substances using the OWRK method and found that the value of the hydrophilicity index $\sqrt{\gamma_{S}^{+}}+\sqrt{\gamma_{S}^{-}}$ can be an important criterion for measuring the hydrophilicity or hydrophobicity of solid surfaces. The requisite condition for solid surface hydrophobicity in aqueous environments is that the hydrophilicity index exceeds 5 mJ/m^2^, while conversely, it indicates a hydrophilic surface. Previous research has demonstrated that among the various titrants, water–glycerol–diiodomethane, water–formamide–diiodomethane, and water–ethylene glycol–diiodomethane combinations can more accurately reflect the surface energy of the material [54]. Superhydrophobic surfaces can generally be achieved via two primary approaches. The first involves creating microstructures on the surface of hydrophobic materials, while the second entails depositing low-surface-energy substances on rough surfaces. Numerous reports have detailed the development of superhydrophobic materials, and the methods for producing these materials can be categorized as follows: the solidification of molten materials, etching, chemical vapor deposition, anodic oxidation, sublimated material blending, phase separation, and templating [54]. Drag reduction experiments conducted on superhydrophobic surfaces revealed that these prepared surfaces can reduce the resistance by approximately 30–40%. However, as the velocity increases, the drag reduction rate decreases to around 14%, primarily due to the influence of surface roughness, which supersedes the impact of surface substances and is the most pivotal factor affecting surface hydrophobicity. Adopting a biomimetic strategy, Ware et al. [57] emulated the lubricating mechanism observed in pitcher plants. They achieved this by creating nanostructured, wrinkled surfaces using materials such as Teflon, polystyrene, and poly(4-vinylpyridine) and subsequently infusing these surfaces with silicone oil. The infused surfaces effectively impeded the proliferation of *Pseudoalteromonas* spp. bacteria, with inhibition rates reaching up to 99%, and they alleviated the adhesion of algae during a 7-week field test. Nonetheless, it was noted that algal attachment increased gradually as the silicone oil gradually depleted over time. Slippery liquid-infused surfaces (SLIS) possess potential applications owing to their unique liquid repellency and self-cleaning properties. However, they are complicated to prepare, offer a low level of protection, and are not durable. Yang et al. [58] reported a facile and efficient liquid infusion method to construct SLISs by altering the heterogeneous microstructure on the SiOx layer of the coatings with direct UV irradiation. The antifouling coatings showed outstanding antifouling properties against various pollutants. The synergistic strategy between lubricant infusion and the coating’s heterogeneous microstructure introduces new possibilities for fabricating promising antifouling coatings.

## 5. Antifouling Strategies Based on Marine Biological Secretions

The two primary methods significantly impact the surface of marine metals. In addition to the aforementioned surface microstructure, surface substances are crucial factors that influence fouling. The surfaces of marine organisms are rich in complex organic compounds, which offer protection against fouling organisms. Many organic compounds have successfully been extracted, such as lobster myosin, water-based extracts, dichlorocarbamine sesquitene, and guaiacoid sesquitene from coral and sponges, effectively inhibiting specific fouling organisms [28]. Nonetheless, due to their limited spectral properties, these compounds are significantly challenging to develop. Some organisms like crabs produce enzymes capable of breaking down the adhesion proteins in fouling organisms, such as algae and barnacles, which helps prevent these fouling organisms from adhering and solidifying. This strategy leverages biological antagonism principles to reduce the fouling organisms’ adhesion [59]. Furthermore, dolphin skin secretes a viscous protein that minimizes seawater resistance [60].

The initial stage of ship fouling involves the adhesion of bacteria and fungi, so studying the interaction between microorganisms and surfaces is necessary and effective. Once the microorganisms adhere to the hull, they produce adhesive proteins, and many natural proteins repel each other. Scientists can capitalize on this property to acquire specific natural proteins or prepare them in a laboratory, naturally deterring the adhesive proteins secreted by microorganisms and preventing the attachment of larger fouling organisms without harming the marine environment. This approach is an environmentally friendly method for ship antifouling and drag reduction. Lots of research has been conducted on natural antifoulants isolated from terrestrial plants and marine organisms. Liu et al. [61] successfully isolated four distinct cardenolides, which belong to the steroid class, from *N. oleander* plants. These isolated compounds were identified as odoroside A, digitoxigenin, oleandrin, and odoroside H. Remarkably, these compounds exhibited exceptional inhibitory properties against barnacle settlement. Additionally, when assessed for their impact on a non-target organism, *Artemia salina* L., they revealed moderate-to-low-level toxicity. This broad spectrum of activity makes these natural antifoulants derived from *N. oleander* particularly promising for large-scale applications, especially considering the widespread presence of *N. oleander* in many subtropical and tropical regions worldwide, where it can be readily harvested. The work of Zhang et al. [62] synthesized 26 subergorgic acid derivatives of this acid. The inhibitory effect of these derivatives against the settlement of *B. amphitrite* was mainly attributed to the presence of double bonds and ketone carbonyl functional groups. Furthermore, the introduction of benzyl esters improved the antifouling properties, while introducing a methylene chain into subergorgic acid reduced the antifouling potency. *Dictyophora indusiata* polysaccharide was applied as a natural antifouling surface modifier to prepare the surface coating for marine antifouling. Three DIP coatings were prepared and the antifouling and anticorrosion behavior of the coatings were examined. The results showed that the DIP coatings had excellent antifouling properties, which could effectively prevent the adhesion of *Chlorella* and the attachment of water-based and oily stains on the surface [63].

Chen et al. [64] isolated indole alkaloids from a fungus (*Eurotium* sp.) derived from a gorgonian and assessed their antifouling properties against the larval settlement of the barnacle *B. amphitrite*. While it is important to note that most of these studies have been conducted in controlled laboratory settings, they underscore the potential of natural antifoulants in the development of effective yet non-toxic compounds for antifouling applications. Furthermore, conducting more extensive research into the mechanisms and structure-activity relationships of natural antifoulants will offer valuable insights for the targeted development of synthetic antifouling agents and innovative antifouling strategies. Currently, numerous peptides and proteins with a strong affinity for metals have been discovered. Most of them have been obtained via phage display studies and have been applied in nanoscience and biotechnology [65]. A peptide was employed as the target molecule to facilitate the creation of a hydrophobic/hydrophilic metal surface on stainless steel. This peptide modification effectively endowed the steel surface with protection against biofouling [66]. In the context of water purification processes such as desalination, antimicrobial peptides (AMPs) were covalently linked to membrane surfaces. This was achieved by incorporating the photoreactive amino acid, 3-(4-benzoylphenyl)alanine, into the AMP sequence, enabling attachment to the surface via filtration. This modification notably increased the surface’s hydrophilicity and induced moderate alterations in the membrane’s performance. Furthermore, this modification resulted in a 55% reduction in bacterial viability and inhibited biofilm formation when exposed to *Pseudomonas aeruginosa* [67]. There are numerous natural antibacterial peptides and proteins, and it is possible to choose ones that strongly interact and bind with metal, creating antibacterial materials. Some examples of natural antibacterial peptides include Temporin B secreted by the glandular granules of European red frogs, analogues of Royal Jellein I (RJI-C) secreted by the upper and lower pharynx of bees, and extracts from butterfly sea urchins [68].

Numerous extensive studies have been conducted, involving the indirect modification of antimicrobial peptides (AMPs) for antifouling purposes on submerged surfaces utilizing various support structures. These modifications have been shown to enhance their impact on biofilm formation. Cao et al. [69] isolated and extracted a macrocyclic oligopeptide from the traditional Chinese medicine *Viola yedoensis* and bound it to the surface of stainless steel, yielding a new antifouling surface. Other antifouling experiments demonstrated robust antifouling surfaces. Single low-surface-energy antifouling methods have limited efficacy. Nisin was effectively attached to a glass microstructural surface via dopamine, demonstrating strong antiadhesive and antibiofilm properties against the alga *Phaeodactylum tricornutum* and the bacterium *Bacillus* sp. [70]. In another investigation, a combinatorial surface modification technique involving dopamine and two synthetic peptides was applied to a 304 stainless steel surface. The resulting modified surface exhibited significantly enhanced antibiofilm effects and antibacterial activity against Staphylococcus aureus compared to those of the untreated and dopamine-only surfaces [71]. Two additional studies employed Magainin II (MAG II) and DA for the creation of antibacterial surfaces. One involved grafting MAG II onto a DA-modified 304 stainless steel surface (SS-DA-MAG). The other entailed grafting a combination of MAG II and DA onto the surface (SS-MAG-DA). The examination of film thickness and surface topology confirmed the successful immobilization of both grafts on the surface. Notably, both modified surfaces displayed robust antibacterial capabilities against *V. natriegens* in 4 weeks, with the physicochemical and antibacterial properties of SS-MAG-DA significantly surpassing those of the SS-DA-MAG-modified surface [72]. A commonly employed modification approach involves using various polymers in conjunction with peptides to modify surfaces, leading to the development of antibacterial coatings. In a separate study, a range of hydrophilic coatings, including zwitterionic, neutral, and positively charged poly(ethylene glycol) (PEG) polymers, were grafted along with covalently attached MAG II peptides onto membrane surfaces. All the AMP-modified membranes demonstrated the effective inhibition of biofilm formation [73]. PDMS- and PEO-based block copolymer coatings were functionalized with non-natural oligopeptide and oligopeptoid side chains. The non-hydrogen bonding peptoid backbone played a primary role in minimizing their adhesion strength to the modified surfaces. Consequently, the adhesion strength and initial attachment properties of *N. incerta* were significantly reduced on surfaces containing peptoid-PEO coatings, underscoring the crucial role of peptide-polymer conjugation in marine antifouling applications [74]. Combining the surface microstructure and low-surface-energy substance modifications to achieve synergistic antifouling surfaces holds significant application potential and research prospects. Xie et al. [75] prepared surfaces with specific morphologies and modified them with various antibacterial substances to create an environmentally friendly and rapidly producible antifouling surface.

## 6. Conclusions

Marine fouling organisms present a significant threat to ship navigation and various submerged surfaces. Low-surface-energy materials offer non-toxic, environmentally friendly, and optically clear marine antifouling coatings, providing distinct advantages over the alternative coatings. Both domestic and international research on these materials has shown significant progress. The research results reveal that numerous factors influence ship antifouling and drag reduction. However, many studies on the impact of different surface morphologies on antifouling and drag reduction remain unresolved. Therefore, further efforts are required from scientists to investigate the mechanisms through which surface morphologies affect ship antifouling and drag reduction, as well as the practical feasibility of implementing these solutions on ships. Future efforts should prioritize the creation of synergistic coatings that amalgamate the biofouling control properties derived from various strategies. However, it is imperative to continually consider multiple factors, such as friendliness to the environment, cost-effectiveness, scalability, and applicability, in diverse marine conditions in the future. We anticipate that the comprehensiveness of this review, coupled with its specific emphasis on marine applications, will serve as a catalyst, inspiring scientists and engineers across various multidisciplinary research fields to develop environmentally friendly and efficient biofouling control coatings and technologies, ultimately contributing to the sustainability of the marine industry.

## Figures and Tables

**Figure 1 biomimetics-08-00502-f001:**
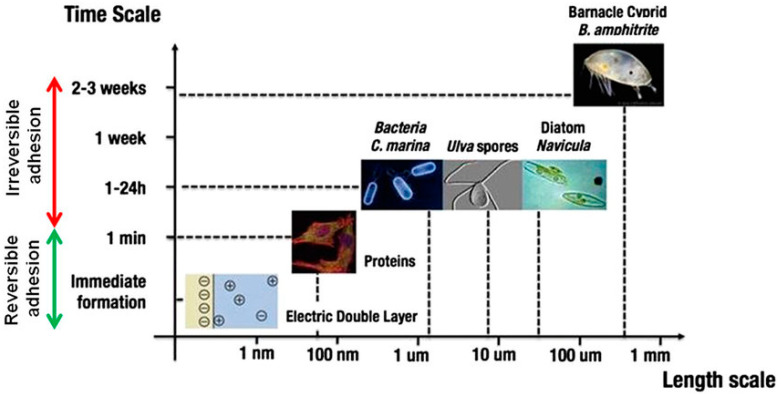
Flow chart of hull fouling and major fouling organisms.

## Data Availability

Not applicable.

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
