# Peer review of "Research Progress on Low-Surface-Energy Antifouling Coatings for Ship Hulls: A Review"

_biomimetics, 2023, doi:10.3390/biomimetics8060502_

Round 1
Reviewer 1 Report
The Manuscript titled: “Research progress in low surface energy antifouling coatings for ship hulls: a review”, by Cao & Cao, takes into account the relevant issue represented by biofouling, in particular related to its effect on ship’s hulls. The subject is for sure of relevant interest for the scientific community working in both new coating development and coastal area management and protection. However, the manuscript suffers from several drawbacks which make it not suitable for publication in MDPI – Biomimetics.
The language is in certain parts not clear, making the manuscript difficult to read, a complete revision by an English native speaker is more than suggested. The subject is treated with superficiality, without adding any novelty to the field. Some statements are out of date and other are incorrect (e.g. line 77: is it not true that all heavy metals are banned, copper and zinc are still widely used in commercial paints). Figures are not useful to the reading and their caption doesn’t help in better understanding (e.g. fig 4 is completely not understandable without reading the paper from which is taken.
The cited literature is quite extensive, however, again, almost half of cited works are more than 10 years’ old. This doesn’t mean that old papers are obsolete, but only that the novelty of the current paper is quite low.
In my own opinion the paper should be deeply revised, first to making it more readable, and then adding some feature, such as metadata analysis or a more critic evaluation, in order to make it more useful for the scientific community.
As already mention in the section before, the manuscript needs an extensive review by an English native speaker.
Author Response
The corresponding modifications in the manuscript are marked in red.
- The language is in certain parts not clear, making the manuscript difficult to read, a complete revision by an English native speaker is more than suggested.
Response: Thank you for your good suggestion. We have revised the entire manuscript in English and have proofread it with a native English speaker.
- Some statements are out of date and other are incorrect (e.g. line 77: is it not true that all heavy metals are banned, copper and zinc are still widely used in commercial paints). Figures are not useful to the reading and their caption doesn’t help in better understanding (e.g. fig 4 is completely not understandable without reading the paper from which is taken.
Response: Thank you for your good suggestion. We have made modifications to inaccurate statements. We agree with the reviewer's opinion that the relevant figures in the manuscript do not effectively demonstrate the purpose of the authors' expression. Therefore, we have deleted the relevant images in the revised manuscript.
- The cited literature is quite extensive, however, again, almost half of cited works are more than 10 years’ old.
Response: Thank you for your good suggestion. We have added and updated references to the revised manuscript and the added and updated references are research results from the past 5 years.
- In my own opinion the paper should be deeply revised, first to making it more readable, and then adding some feature, such as metadata analysis or a more critic evaluation, in order to make it more useful for the scientific community.
Response: Thank you for your good suggestion. We have already deeply revised the manuscript. We updated and added many relevant references firstly, then added necessary data and critical discussion information, and finally did proofreading for the entire manuscript.
Reviewer 2 Report
In the present manuscript the authors present a review on the research progress in low surface energy antifouling coatings for ship hulls. This review is well performed and includes all the necessary recent citations. I recommend publication in its present form.
Minor editing of English language required.
Author Response
- Minor editing of English language required.
Response: Thank you for your suggestion. Thank you for your suggestion. We have revised the English throughout.
Reviewer 3 Report
The paper focused on the research progress of low surface energy antifouling coatings such as organic silicon and organic fluorine for antifouling and drag reduction of ship hulls, and the antifouling mechanism of silicon-based and fluorine-based low surface energy antifouling coatings are explored, their advantages and disadvantages in applications are explained. Overall, this is an interesting paper. Some questions should be illustrated before publishing.
(1) How the antifouling performance changes after long-term aging such as one week, two weeks and one month? More discussing should be added.
(2) The quality of figure 1 is not as good.
(3) The conclusions should provoide more directions for future research.
(4) Some Refs could be cited. eg.Friction (2023). https://doi.org/10.1007/s40544-023-0797-8; https://doi.org/10.1016/j.surfcoat.2022.129124
Moderate editing of English language required
Author Response
The corresponding modifications in the manuscript are marked in blue.
- How the antifouling performance changes after long-term aging such as one week, two weeks and one month? More discussing should be added.
Response: Thank you for your good suggestion. More relevant discussions have been added to the revised manuscript. (line 118-125; line 228-232; line 373-379; line 468-476)
- The quality of figure 1 is not as good.
Response: Thank you for your good suggestion. We have replaced the clear image.
- The conclusions should provoide more directions for future research.
Response: Thank you for your good suggestion. We have already added more directions for future research in the discussion of the revised manuscript.
- Some Refs could be cited. eg. Friction (2023). https://doi.org/10.1007/s40544-023-0797-8; https://doi.org/10.1016/j.surfcoat.2022.129124
Response: Thank you for your good suggestion. We have added and updated references to the revised manuscript, including the two references you mentioned, which are references 58 and 63, respectively.
Round 2
Reviewer 1 Report
The Manuscript was surely improved in relevance by the author's revisions, by removing wrong information and adding more references, however really few was done to differentiate this paper from all other reviews present in the relative literature. In my opinion, after a deep language revision (see following section), the choice is in the Editor hands, in deciding if she/he is interested in a simple collection of information without nothing different from other older reviews on the same subject.
Several grammar errors, in both use of words and period construction, are still present in the manuscript. This makes me doubting in a real revision by an English native speaker. I suggest, again, a really deep revision by an English native speaker.
Author Response
Several grammar errors, in both use of words and period construction, are still present in the manuscript. I suggest, again, a really deep revision by an English native speaker.
Response: Thank you for your good suggestion. We have conducted full text editing and proofreading on the MDPI official language editing website.
Reviewer 3 Report
The revision made by the author is clear and can be considered acceptable.
Author Response
The revision made by the author is clear and can be considered acceptable.
Response: Thank you for your time and effort.